# Black Soldier Fly (*Hermetia illucens*) Frass on Sweet-Potato (*Ipomea batatas*) Slip Production with Aquaponics

Nicholas Romano [1,*], Carl Webster [2], Surjya Narayan Datta [3], Gde Sasmita Julyantoro Pande [4], Hayden Fischer [5], Amit Kumar Sinha [5], George Huskey [2], Steven D. Rawles [2] and Shaun Francis [6]

1  Virginia Cooperative Extension, College of Agriculture, Virginia State University, Petersburg, VA 23806, USA
2  USDA-ARS Harry K. Dupree Stuttgart National Aquaculture Research Center, Stuttgart, AR 72160, USA
3  Department of Fisheries Resource Management, College of Fisheries,
   Guru Angad Dev Veterinary and Animal Sciences University, Ludhiana 141004, India
4  Department of Aquatic Resource Management, Faculty of Marine Science and Fisheries, Udayana University,
   Kuta Selatan 80361, Bali, Indonesia
5  Department of Aquaculture & Fisheries, University of Arkansas at Pine Bluff, 1200 N University Dr.,
   Pine Bluff, AR 71601, USA
6  1890 Cooperative Extension Program, University of Arkansas at Pine Bluff, 1200 N University Dr.,
   Pine Bluff, AR 71601, USA
*  Correspondence: nromano@vsu.edu; Tel.: +1-804-524-5620

**Abstract:** Nutrient supplementations are often added to aquaponic systems to optimize plant production, and black soldier fly larvae frass is a promising organic fertilizer. However, the mineral composition of the frass is substantially influenced by the initial substrate. In an 8-week study, sweetpotato slips were cultured at commercial stocking densities in an aquaponic system which received weekly additions of either BSFL frass made from high-nitrogen expired fish diets or low-nitrogen fruits/vegetables. The sweetpotato slips ($\geq$8 nodes) were harvested weekly. Despite differences in the mineral composition between the frass types, the water quality as well as slip production/sugar content were unaffected by frass type. The results indicate that a wide array of substrates may be suitable for producing black soldier fly larvae frass as a fertilizer in aquaponic systems. Lastly, aquaponics is a viable system to commercially produce sweetpotato slips.

**Keywords:** sweetpotato slips; insect frass; black soldier fly larvae; organic fertilizer

## 1. Introduction

Sweetpotatoes (*Ipomea batatas*) are the sixth-most produced crop in the world and are increasingly being recognized as a 'super food' due to their high content of health-promoting carotenoids, vitamins, and minerals [1,2]. The storage roots are grown in soil and while the roots can be used again as planting material, there is a point when this practice is no longer viable due to the accumulation of viruses and mutations that limit their production [3]. In response, virus-indexed planting material, known as 'slips', are produced via apical meristem culture and are then vegetatively multiplied [4]. These first-generation slips are often cultured in greenhouses in order to accelerate their growth and thus allow a longer growing season for their later storage root production. Sweetpotato farmers rely on obtaining a sufficient amount of slips and thus, sweetpotato slip production is itself an industry. Nevertheless, shortages are still common, and it has been suggested that the nutritional requirements of sweetpotato slips are not fully known and further research on the most appropriate fertilizer is an area that could be improved to optimize production [5,6].

One promising method to grow slips under controlled conditions is with aquaponics, which is the symbiotic fusion of aquaculture with hydroponics [7]. In aquaponics, the waste excreted from the fish acts as nutrients for plants to enhance sustainability and

profitability due to the production of two marketable food items. Recently, it was shown that sweetpotato slips grown in an aquaponic system led to a five-fold and thousand-fold increase in the amount of slips and total weight, respectively, compared to those grown in soil [8]. The constant supply of water and nutrients, particularly dissolved nitrogen, were suggested as the contributors to this result. However, it is common practice to supplement additional minerals, most notably iron (Fe), calcium (Ca), and potassium (K), to the system to enhance plant production. Aquaponic supplements are often added in synthetic forms, such as potash, rock phosphate, Epsom salt (magnesium sulfate), and CalMag (calcium magnesium), but farmers interested in organic farming could be interested in an organic fertilizer that contains a variety of essential nutrients.

A by-product of insect farming is 'frass', which is the mineral-rich excrement of insects [9]. In particular, BSFL frass is relatively high in essential minerals compared to the frass of other edible insect species [10], while the accumulated chitin in the frass may also act as a plant prebiotic [11]. The use of BSFL frass on terrestrial plants has, in some cases, exceeded production compared to synthetic fertilizers [10,12,13]. In an aquaponic context, adding BSFL frass tea enhanced the sugar content in sweet banana peppers and the manganese content in sweetpotato slips grown aquaponically but had no influence on the overall production [14]. However, the growth of collard greens significantly increased when higher amounts of BSFL frass were added in an aquaponic system [15]. While the amount of BSFL frass additions likely plays a role in plant growth, the type of the initial substrate used to produce the frass may also be a factor. This is because the initial substrate greatly influences the mineral composition of the BSFL frass [16]. For example, the use of vegetable waste led to significantly higher phosphorus and potassium content in BSFL frass compared to either fruit waste or starches [17].

Nile tilapia (*Oreochromis niloticus*) is a tropical cichlid native to the Middle East (Jordan, Egypt, and Israel) and parts of Africa and is the third-most cultured fish in the world [18]. The fish has many highly desirable culture traits which include rapid growth rate, excellent flesh taste and quality, resistance to numerous diseases, ability to reproduce easily in captivity, possession of dietary requirements on the lower end of the food chain (herbivorous/omnivorous), and the ability to tolerate varied environmental and production conditions [19]. As Nile tilapia are a popular choice for use in aquaponic systems, growth and survival of the fish can serve as a benchmark when conducting aquaponic research.

The aim of this study was to compare the growth and mineral composition of sweetpotato slips under commercial stocking conditions in an aquaponic system receiving supplementations of BSFL frass produced with expired fish diets (EFD) or from fruits/vegetables (FV). It was hypothesized that the different frass types would have a different elemental composition and thus additions of these to an aquaponic system would influence the production and/or composition of the sweetpotato slips.

## 2. Materials and Methods

### 2.1. Source of Plants, Fish, and BSFL Frass

Virus-indexed sweetpotato slips were produced and provided by the Agriculture Department at the University of Arkansas at Pine Bluff (UAPB), which had at least 7 nodes. The all-male tilapia used in this study were purchased from AZGardens and upon arriving at UAPB, these were kept in a 1000 L acclimation tank. The fish were fed once daily with a floating commercial pellet (32% protein) designed for tilapia. The BSFL frass were produced in the lab according to Fischer et al. [20]. Briefly, the eggs of BSFL were placed on top of spoiled fish feeds designed for catfish (Rangen; 32% protein) or a combination of fruits (orange peels, banana peels, apple cores, and strawberries) and vegetables (sweetpotato and peas). The approximate composition of the SF and FV was measured using the standard Association of Official Analytical Chemists [21] methods and results are presented in Table 1. Hereafter, the frass made from spoiled feeds or fruits/vegetables will be referred to as SF frass and FV frass, respectively.

**Table 1.** Approximate composition (% dry matter) of expired fish diets (EFD) and fruit/vegetables (FV) that were provided to the black soldier fly larvae.

|  | **EFD** | **FV** |
|---|---|---|
| Moisture | 5.23 | 85.95 |
| Crude protein | 32.13 | 9.32 |
| Crude lipid | 8.47 | 3.28 |
| Crude ash | 7.64 | 14.52 |
| Crude fiber | 4.31 | 9.27 |

The BSFL converted these into frass, which took approximately three weeks. The frass was then dried in a forced air oven (Despatch; LBB Series 2–12-3, Illinois Tool Works, Inc., Minneapolis, MN, USA) at 100 °C for two days and then ground into a fine powder with a hammer mill. Frass nitrogen was measured using a Leco N analyzer while the mineral content was measured at the Fayetteville Agricultural Diagnostic Laboratory at the University of Arkansas with inductively coupled plasma (ICP) analysis. The results are shown in Table 2.

**Table 2.** Nutrient composition of black soldier fly larvae frass produced with expired fish diets (EFD frass) or fruits/vegetable (FV frass).

|  | **%** | | | | | | **mg/kg** | | | | | |
|---|---|---|---|---|---|---|---|---|---|---|---|---|
|  | **N** | **P** | **K** | **Ca** | **Mg** | **S** | **Na** | **Fe** | **Mn** | **Zn** | **Cu** | **B** |
| EFD frass | 4.64 | 2.54 | 2.95 | 5.28 | 0.44 | 0.75 | 13,561 | 463 | 87 | 200 | 30.7 | 23 |
| FV frass | 3.37 | 1.16 | 4.12 | 6.38 | 0.38 | 0.50 | 11,815 | 295 | 63 | 104 | 22.2 | 33 |

### 2.2. Aquaponic Systems and Experimental Design

There were a total of six identical aquaponic systems (5110 L capacity) that are described in detail in Romano et al. [14]. A total of 40 tilapia (initial weight of 45.7 g) were added into each aquaponic system and fed twice daily to apparent satiation with commercial floating feeds (Rangen; 32% protein). Each tank received gentle aeration with an air stone and the tanks were covered with netting to provide shade (to minimize algae growth) and prevent any escapees. The amount of food provided to each system was recorded.

After one week of feeding the fish, a total of 200 sweetpotato slips were planted in each of the aquaponic media beds and each slip was spaced 2.5 cm (or 1 inch) apart from each other in media bed (145 cm × 75 cm) filled with expanded lava rock. This stocking density is the same used in commercial settings [5]. After adding the slips, a total of 10 g of SF frass or FV frass were sprinkled on top of the media bed containing the slips. This was performed to potentially encourage frass mineralization but also because, based on past experience, adding frass to the sump encouraged filamentous algae growth. Every week, waste that settled in the sump was siphoned out followed by adding 5 mL of iron chelate (Iron-gluconate; SEACHEM Flourish, Root 98 Warehouse, Lakeland, Florida). No buffers were added to adjust pH.

### 2.3. Water Quality Analysis

The ammonia-N, nitrite-N, and nitrate-N levels were measured with an API master test kit once a week. The water temperature, dissolved oxygen and pH were measured with a digital multimeter probe (YSI Professional Plus). On week 2, 4, 6, and 8, a water sample was collected from the sump for later mineral analysis and stored at −20 °C. The minerals were measured with a flame atomic absorption spectrophotometer (AAS, iCE 3000 series, Thermo Scientific, Santa Clara, CA, USA) with deuterium lamp background correction. Calibrations were made using single element standard solutions (CPI International, Santa Rosa, CA, USA). However, for phosphorus (P), the persulfate digestion method (HACH method 8190) was used because the concentrations were too low for the AAS.

## 2.4. Aquaponic Sampling

By week 2 of adding the cuttings, the majority grew to be considered a slip (≥6 nodes) and were harvested to allow at least 2 nodes remaining in the media bed. The total biomass of all the slips were weighed among the treatments while 40 were used to measure the total length, number of nodes, and stem diameter. These were then placed in a zip lock plastic bag and stored at −20 °C for later mineral and sugar analysis. After 8 weeks, the slips were harvested 7 times in total and all the remaining slips were counted to determine the overall survival. The tilapia were also harvested after 8 weeks, counted, and were individually measured for weight and length using a digital scale and metric ruler, respectively.

## 2.5. Mineral and Sugar Analysis

For mineral analysis, the sweetpotato leaves from each replicate were oven-dried at 60 °C for 24 h then digested in a heat block (Environmental Express, Charleston, SC, USA) at 115 °C for 30 min in 4.0 mL trace metal-grade $HNO_3$ (69%; Sigma-Aldrich). After digestion, 0.1 mL of $H_2O_2$ (30%) was added and then 40 mL of Milli-Q water was added. Samples were measured for iron, calcium, zinc, magnesium, and manganese using a flame atomic absorption spectrophotometer (AAS) (iCE 3000 series, Thermo Scientific, Santa Clara, CA, USA), while phosphorus was measured using the persulfate digestion method (HACH method 8190) as described above. Each replicate sample was measured in triplicate.

For sugar analysis, total sugar was estimated from 100 mg of sweetpotato leaves from each replicate that were ground in liquid nitrogen using a mortar and pestle. Ground samples were transferred to 1 mL 100% acetone and kept overnight at 4 °C. Samples were centrifuged and the residue was repeatedly washed with hot 80% ethanol to remove all traces of soluble sugars. This filtrate was used for the determination of soluble sugars while the residue/pellet was used for the determination of the insoluble sugar content. To the residue, 2 mL of 0.2 N $H_2SO_4$ was added, followed by heating at 100 °C in a water bath for 30 min, which was then centrifuged and the supernatant was collected. Anthrone reagent (150 µL) was added to each microplate well containing 50 mL of glucose standard solutions, blanks, and samples (soluble and insoluble sugars). Plates were then placed for 10 min at 4 °C and then incubated for 20 min at 100 °C. A cooling step for 20 min at room temperature was completed before reading absorbance at 620 nm triplicate in a microplate reader (PowerWave XS, BioTek Instruments, Winooski, VT, USA). A standard curve was obtained with different concentrations of glucose. Each replicate sample was measured in triplicate.

## 2.6. Statistical Analysis

Sweetpotato slip composition and performance in response to frass type (SF vs. FV) and harvest period (first versus final composition, or slip performance among harvests 1–7) were analyzed by factorial mixed model (MIXED) analysis of variance (SAS version 9.4, SAS Institute, Inc., Cary, NC, USA). Aquaponic system water quality and tilapia growth performance in response to frass type were analyzed by one-way mixed model analysis of variance. All response values were natural log-transformed prior to analyses. Differences among response means were considered significant at $p \leq 0.05$.

## 3. Results

### 3.1. Frass Characteristics

The approximate compositions of the two initial substrates and subsequent frass are presented in Table 1. The frass made from the expired fish diet (EFD) contained 30.79% protein, 7.69% lipid, and 24.83% ash. This is in contrast to the frass made from fruits/vegetables (FV) which contained 22.73% protein, 4.03% lipid, and 31.16% ash.

The analyzed mineral compositions of each BSFL frass is shown in Table 2. There were differences in the mineral compositions for each frass with FV frass having higher levels of N, P, Mg, S, Na, Fe, Mn, Zn, and Cu compared to EFD frass, while EFD frass had higher levels of K, Ca, and B than FV frass.

### 3.2. Water Quality and Chemistry

The addition of the two types of BSFL frass to media beds led to only one difference in measured water quality parameters and all were within acceptable limits for fish. The aquaponic system water quality was optimum for tilapia culture and did not differ markedly in response to frass type (Table 3). Averages during the study were as follows: temperature, 27.4 °C; dissolved oxygen, 5.31 mg/L; hardness, 47.4 mg/L; total ammonia nitrogen (TAN), 0.33 mg/L; nitrite, 0.05 mg/L; nitrate, 42.08 mg/L. Only the pH was statistically higher in beds receiving the FV frass (7.54) compared to the SF frass (7.49), which was not biologically significant.

**Table 3.** Mean (±SE) water quality parameters in an aquaponic system receiving frass by black soldier fly (*Hermetia illucens*) larvae fed expired fish feeds (EFD) or fruits/vegetables (FV) over 8 weeks.

| Parameter | EFD | FV |
|---|---|---|
| Temperature (°C) | 27.3 ± 0.05 | 27.5 ± 0.09 |
| Dissolved oxygen (mg/L) | 5.14 ± 0.33 | 5.47 ± 0.01 |
| pH | 7.49 ± 0.01 [b] | 7.54 ± 0.01 [a] |
| Hardness (mg/L) | 45.5 ± 0.14 | 49.3 ± 0.27 |
| TAN [1] (mg/L) | 0.34 ± 0.00 | 0.31 ± 0.02 |
| Nitrite (mg/L) | 0.08 ± 0.04 | 0.03 ± 0.00 |
| Nitrate (mg/L) | 42.08 ± 0.41 | 42.08 ± 0.12 |

[1] Total ammonia nitrogen (TAN). Different superscripted letters indicate significant difference ($p < 0.05$).

### 3.3. Fish Growth

The tilapia growth performance was typical for this species and initial size grown in optimum conditions, and there were no significant differences between frass types (Table 4). The averages for length (cm), final weight (g), weight gain (%), specific growth rate (SGR; %/day), feed intake (g diet/fish), and feed conversion ratio (FCR) were: 21.09 cm, 193.1 g, 147.35 g, 322.2%, 2.95%/day, 162.15 g diet/fish, and 1.11, respectively.

**Table 4.** Growth performance of Nile tilapia (*n* = 3) in an aquaponics system receiving frass by black soldier fly (*Hermetia illucens*) larvae fed expired fish feeds (EFD) or fruits/vegetables (FV) after 8 weeks.

| Response | EFD | FV |
|---|---|---|
| Length (cm) | 21.23 ± 0.19 | 20.95 ± 0.29 |
| Final weight (g) | 197.3 ± 4.6 | 188.9 ± 8.0 |
| Weight gain (%) [1] | 332.6 ± 13.8 | 311.8 ± 13.4 |
| SGR [2] | 3.00 ± 0.56 | 2.89 ± 0.42 |
| Feed intake (g/fish) | 164.6 ± 5.8 | 159.7 ± 2.3 |
| FCR [3] | 1.09 ± 0.06 | 1.12 ± 0.04 |

[1] Percent gain from initial weight. [2] Specific growth rate (SGR) = (ln final weight—ln initial weight)/time. [3] Feed conversion ratio (FCR) = g dry feed fed/g gained.

### 3.4. Sweetpotato Slip Production and Mineral Composition

The sweetpotato slip production and quality were unaffected by the frass type but differed among partial harvests (Table 5). Generally, the slip diameter decreased, whereas the node length and number of nodes per slips increased between the initial (#1) and final harvests (#7). The slip length and weight differed by harvest with no discernible pattern.

The sweetpotato slip mineral and sugar composition was unaffected by the frass type but differed between the initial and final harvests (Table 6). The concentrations of Fe, Mn, and Zn increased at the final harvest from the initial content, whereas the Ca and Na concentrations decreased at the final harvest. The concentrations of Mg, P, and K in the slips were unchanged between the initial and final harvests and were unaffected by frass type. The soluble and insoluble sugars decreased significantly by three-fold and six-fold, respectively, from the initial concentrations.

**Table 5.** Mean (±SE) production parameters of sweetpotato slips (*n* = 3) grown in an aquaponics system with Nile tilapia for 8 weeks when black soldier fly (*Hermetia illucens*) larvae frass was produced from expired fish feeds (EFD) or fruits/vegetables (F/V). Main effects of least squares means with different letters are significantly different (*p* < 0.05).

| Treatments | | Response Variables | | | | |
|---|---|---|---|---|---|---|
| Frass | Harvest | Length (cm) | Weight (g) | Diameter (mm) | Nodes/Slip | Nodes/Length |
| EFD | 1 | 31.15 | 912.7 | 4.37 | 7.69 | 0.263 |
| | 2 | 27.75 | 259.7 | 3.21 | 8.88 | 0.343 |
| | 3 | 50.18 | 1029.7 | 3.10 | 9.35 | 0.213 |
| | 4 | 29.52 | 381.7 | 3.06 | 8.69 | 0.333 |
| | 5 | 39.89 | 644.6 | 2.97 | 10.33 | 0.300 |
| | 6 | 22.23 | 205.1 | 2.90 | 8.78 | 0.420 |
| | 7 | 31.03 | 633.3 | 2.88 | 11.14 | 0.397 |
| FV | 1 | 30.28 | 758.7 | 4.24 | 7.57 | 0.260 |
| | 2 | 31.81 | 486.6 | 3.30 | 8.88 | 0.310 |
| | 3 | 48.03 | 1102.9 | 3.11 | 9.49 | 0.213 |
| | 4 | 30.30 | 413.7 | 3.24 | 8.74 | 0.310 |
| | 5 | 42.85 | 649.9 | 2.98 | 10.08 | 0.273 |
| | 6 | 29.68 | 299.3 | 2.89 | 9.50 | 0.367 |
| | 7 | 31.41 | 828.5 | 2.95 | 11.59 | 0.393 |
| Pooled SE | | 4.44 | 138.7 | 0.12 | 0.45 | 0.026 |
| Main effects of means | | | | | | |
| EFD | | 33.11 | 581.0 | 3.21 | 9.27 | 0.324 |
| FV | | 34.91 | 648.5 | 3.24 | 9.40 | 0.304 |
| | 1 | 30.72 [bc] | 835.7 [a] | 4.31 [a] | 7.63 [d] | 0.262 [bc] |
| | 2 | 29.78 [c] | 373.2 [c] | 3.25 [b] | 8.88 [bc] | 0.327 [ab] |
| | 3 | 49.11 [a] | 1066.3 [a] | 3.10 [bc] | 9.40 [bc] | 0.213 [c] |
| | 4 | 29.91 [c] | 397.7 [bc] | 3.15 [bc] | 8.72 [cd] | 0.322 [b] |
| | 5 | 41.37 [ab] | 647.3 [ab] | 2.97 [bc] | 10.20 [a] | 0.287 [b] |
| | 6 | 25.95 [c] | 252.2 [c] | 2.89 [c] | 9.14 [bc] | 0.393 [a] |
| | 7 | 31.22 [bc] | 730.9 [ab] | 2.91 [c] | 11.36 [a] | 0.395 [a] |
| ANOVA Source, Pr > F | | | | | | |
| Frass | | 0.638 | 0.694 | 0.792 | 0.777 | 0.463 |
| Harvest | | <0.001 | <0.001 | <0.001 | <0.001 | <0.001 |
| F × H | | 0.818 | 0.922 | 0.881 | 0.907 | 0.838 |

**Table 6.** Mean (±SE) mineral (mg/g dry weight) and sugar composition (mg/g dry weight) of sweetpotato slips (*n* = 3) grown in an aquaponics system with Nile tilapia for 8 weeks when black soldier fly (*Hermetia illucens*) larvae frass was produced from expired fish feeds (EFD) or fruits/vegetables (F/V). Main effects of least squares means with different letters are significantly different (*p* < 0.05).

| Treatments | | Macronutrients (mg/g) | | | | Micronutrients (mg/g) | | | | Sugar (mg/g) | |
|---|---|---|---|---|---|---|---|---|---|---|---|
| Harvest | Frass | P | K | Ca | Mg | Na | Fe | Mn | Zn | Soluble | Insoluble |
| First | EFD | 9.47 | 109.24 | 7.49 | 10.94 | 0.071 | 0.121 | 0.137 | 0.071 | 30.08 | 26.20 |
| | FV | 9.50 | 109.80 | 7.47 | 11.17 | 0.090 | 0.118 | 0.134 | 0.090 | 37.63 | 35.29 |
| Last | EFD | 9.79 | 92.73 | 6.71 | 10.78 | 0.175 | 0.148 | 1.500 | 0.175 | 8.83 | 5.55 |
| | FV | 9.17 | 90.95 | 6.89 | 10.46 | 0.187 | 0.142 | 1.499 | 0.187 | 13.55 | 6.01 |
| Pooled SE | | 0.385 | 8.36 | 0.165 | 0.21 | 0.047 | 0.006 | 0.014 | 0.015 | 4.25 | 4.83 |
| First | | 9.48 | 109.52 | 7.48 [a] | 11.05 | 0.080 [b] | 0.120 [b] | 0.135 [b] | 0.080 [b] | 33.86 [a] | 30.75 [a] |
| Last | | 9.48 | 91.84 | 6.80 [b] | 10.62 | 0.181 [a] | 0.145 [a] | 1.499 [a] | 0.181 [a] | 11.19 [b] | 5.78 [b] |
| | EFD | 9.63 | 100.99 | 7.10 | 10.86 | 0.123 | 0.134 | 0.819 | 0.123 | 49.46 | 15.87 |
| | FV | 9.33 | 100.38 | 7.18 | 10.82 | 0.138 | 0.130 | 0.816 | 0.138 | 25.59 | 20.65 |

**Table 6.** *Cont.*

| Treatments | | Macronutrients (mg/g) | | | | Micronutrients (mg/g) | | | | Sugar (mg/g) | |
|---|---|---|---|---|---|---|---|---|---|---|---|
| Harvest | Frass | P | K | Ca | Mg | Na | Fe | Mn | Zn | Soluble | Insoluble |
| ANOVA Source, Pr > F | | | | | | | | | | | |
| Time | | 0.934 | 0.062 | 0.003 | 0.071 | 0.035 | 0.004 | <0.001 | <0.001 | <0.001 | <0.001 |
| Frass | | 0.456 | 0.456 | 0.646 | 0.837 | 0.165 | 0.528 | 0.860 | 0.324 | 0.096 | 0.422 |
| T X F | | 0.427 | 0.752 | 0.558 | 0.893 | 0.537 | 0.826 | 0.893 | 0.774 | 0.757 | 0.506 |

## 4. Discussion

One of the criticisms of aquaponics is that the production of plants is often limited in scale compared to terrestrial farming, where hundreds or thousands of acres of crops can be grown outside and subsequently harvested with tractors and other heavy machinery. Thus, aquaponics is often viewed as being more suitable for growing niche crops and/or farming in an urban environment where space is limited [7]. However, in the case of virus-indexed sweetpotato slips, these are often cultivated anyway under controlled environmental conditions, like greenhouses, to optimize growth before being transplanted outdoors in the soil [5]. Consequently, obtaining a sufficient amount of slips is a bottleneck in the sweetpotato industry and any method to optimize slip growth would help extend the season for storage root production in soil [4].

Aquaponics appears to be a viable method for sweetpotato slip production in which it has been previously shown that sweetpotato slips grew substantially faster in aquaponic conditions compared to those grown in soil [8]. It is known that an abundance and consistent supply of water and nitrogen in aquaponic systems can promote leafy growth in a variety of plants [7], and inhibit storage root production in sweetpotatoes [6,22]. Indeed, under aquaponic conditions, storage root growth was not observed and thus it was suggested that more energy could be diverted to leafy growth [14]. However, the stocking density used in Romano et al. [14] was low (three cuttings in a 145 cm × 75 cm plant culture bed) and not representative of commercial conditions. Thus, a higher stocking density of cuttings was adopted in this study in which a total of 200 cuttings were planted in each of the media beds (145 cm × 75 cm).

After the cuttings were planted in this study, it took about a week before these began to discernibly start growing, which indicates a period of acclimation to the new culture conditions. By the second week, the cuttings were sufficiently long enough to be considered slips because they had at least six nodes; thus, the slips were subsequently harvested by week two of this study. Each week thereafter the slips were harvested another six times (for a total of seven harvests) until the study concluded when the fall season was approaching. Typically, sweetpotato slips grown in soil are harvested between 10 to 14 days to provide a total of nine harvests in a year, first starting in April [6]. In this study, harvesting was up to two-fold faster where it could be possible to harvest up to 18 or more times before the season ends. Additionally, the number of nodes on each slip gradually increased with each harvest, which is considered desirable because more nodes means more planting material for storage root production in the soil.

While nitrogen is generally abundant in aquaponics systems, the most common limiting nutrients include K, Ca, and Fe and these are often added in the forms of potassium bicarbonate, CalMag, and iron chelate, respectively [14,20,23]. However, there are other essential macro- and micro-nutrients that may be at insufficient levels for the optimal growth and well-being of plants in aquaponic systems. In this common scenario, adding a mineral-rich fertilizer consisting of various essential minerals may be effective. It was previously shown that directly adding BSFL frass tea to the water of an aquaponic system had no effect on sweetpotato slip production [14], but when added in a quantity more than two-fold higher this significantly increased collard green growth [15]. More recently, dietary inclusions of BSFL frass at 10% increased the growth of catfish (*Ictalurus punctatus*)

as well as stevia (*Stevia rebaudiana*), and lavender (*Lavaridula angustifolia*) in an aquaponic system [24].

While there appears to be strong indications that adding BSFL frass can provide benefits to aquaponic plants, it is known that the composition of BSFL frass greatly depends on the initial substrate provided. However, to date, the efficacy of BSFL frass made from different substrates has not been compared in aquaponics. The two different types of BSFL frass that were compared in this study were produced with high-nitrogen expired fish diets (EFD) while the other with low-nitrogen fruits/vegetables (FV). The EFD frass did indeed have a higher nitrogen content, but the difference was not as remarkable as the difference between the initial substrates. In terms of the limiting nutrients in aquaponic systems, the EFD frass had less K and Ca, but more Fe than the FV frass. Despite the different mineral composition of the BSFL frass, additions of these different frass types led to no difference in sweetpotato slip production. It could be argued that the amounts added were insufficient to make a difference. While this could be a factor, it is perhaps worthy to note that 10 g of BSFL was added weekly in this study compared to 2.5 g each week which was sufficient to enhance collard green growth in the same system with similar stocking densities of fish [15]. Even though higher amounts were added in this study, the mean ammonia levels did not exceed 0.5 mg/L, while the other water quality parameters were similar.

Among the tested minerals in the water, P was significantly higher in the EFD treatment at week 8. It is tempting to attribute the higher P to the EFD frass having an over two-fold higher P content. However, this would not explain the K water content being significantly lower in both the water and slips from the FV frass treatment at week 6, because the K content was higher in the FV frass. Moreover, the Fe content of the slips was significantly higher in FV frass treatment, despite the FV frass having almost two-fold less Fe. Nevertheless, Fe as well as Mn were consistently at undetectable levels in the water. This was despite the weekly additions of iron chelate (along with the BSFL frass), indicating that Fe was being absorbed by the slips at a faster rate than the inputs of this nutrient. Indeed, this seems to be supported by the increased Fe and Mn (as well as Zn) content of the sweetpotato slips compared to their initial values. It is conceivable that sweetpotato slip production could be further enhanced by ensuring Fe is not limiting and perhaps should be monitored more closely. It is important to point out, however, that chlorosis, which is yellowing of the leaves and a symptom of Fe deficiency, was not observed in this study.

It has been demonstrated that fish grown in an aquaponic system have normal growth and survival compared to traditional production methods. Some of the species successfully grown aquaponically include largemouth bass (*Micropterus salmoides*) [20], channel catfish (*Ictalurus punctatus*) [24], Nile tilapia (*Oreochromis niloticus*) [25], rainbow trout (*Oncorhynchus mykiss*) [26,27], goldfish (*Carassius auratus*) [28], and white shrimp (*Litopenaeus vannamei*) [29]. In this study, the fish growth was acceptable and similar to the production parameters in other reports [30–33] and the tilapia growth was not adversely affected by the frass type.

## 5. Conclusions

Inclusions of BSFL frass, either directly to the water or in aquafeeds, has been previously shown to benefit plant production in an aquaponic system. The results of this study appear to indicate that the initial substrate used to make BSFL frass is not a major factor in sweetpotato slip production. Considering that the farming of BSFL is expected to increase in the coming years, this could also increase the availability of BSF frass as an option to aquaponic farmers, particularly those interested in using an organic fertilizer rather than traditionally relying on synthetic ones. Finally, the production of sweetpotato slips at commercially stocking densities appears to be a viable farming method that may improve slip availability and help extend the duration for storage root production. Further studies on optimizing the BSFL frass dose and potential ways to enhance the Fe and Mn in the BSFL frass may further improve plant production.

**Author Contributions:** Conceptualization, methodology, supervision, and writing original draft preparation, N.R.; formal analysis, S.N.D., G.S.J.P., G.H. and A.K.S.; data curation and final editing, N.R., S.D.R. and C.W.; assistance with conceptualization, H.F.; provide virus-index slips and technical assistance, S.F. All authors have read and agreed to the published version of the manuscript.

**Funding:** This study was funded by 1890's Institutional Global Food and Nutritional Security Grant and a non-assistance Cooperative Agreement (58-6028-2-005). This research was supported, in part, by funds provided by the USDA/ARS CRIS Project 6028-31630-009-00D and is a research component of the USDA-ARS Grand Challenge project entitled "Debugging a new mini livestock commodity: Developing a model of insect production to demonstrate their value as a safe solution for food waste and sustainable fish and livestock production".

**Data Availability Statement:** Data can be made available with reasonable request.

**Acknowledgments:** We would like to thank John Brewer for the technical assistance.

**Conflicts of Interest:** The authors declare no conflict of interest.

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
