# Peer review of "Black Soldier Fly (Hermetia illucens) Frass on Sweet-Potato (Ipomea batatas) Slip Production with Aquaponics"

_horticulturae, doi:10.3390/horticulturae9101088_

Round 1
Reviewer 1 Report
Suggested corrections:
Title is not that clear, however at the least change to:
“Type of black soldier fly (Hermetia illucens) frass on sweetpotato (Ipomea batatas) slip production…”
2. Sweetpotato and sweet potato used interchangeably in the text, use one or the other.
3. Line 103: check for clarity: “…three weeks, the frass was dried…”
4. Clarify what is initially planted, a slip or a cutting (lie 118), as the sentence (line 141) does not make sense: “By week 2 of adding the slips, the majority grew sufficiently long enough to be considered a slip…” (suggest removing enough)
5. Line 171: Not clear what was done in triplicate (same sample read 3 times, or three different samples).
6. Table 5 and 6 are difficult to read. Units for the different parameters need to be included, include a legend to explain the statistics.
7. Minor citation changes:
Remove Clarke et al. 2002 (line 38) (using numbering system i.e., 3)
Remove Tedesco et al. 2023 (line 263)
Author Response
Reviewer #1
Reviewer comment #1: Title is not that clear, however at the least change to:
“Type of black soldier fly (Hermetia illucens) frass on sweetpotato (Ipomea batatas) slip production…”
Response to comment #1: Thank you, the title has now been changed.
Reviewer comment #2: Sweetpotato and sweet potato used interchangeably in the text, use one or the other.
Response to comment #2: Thank you, this has been changed.
Reviewer comment #3: Line 103: check for clarity: “…three weeks, the frass was dried…”
Response to comment #3: Thank you for pointing this out. This has been revised for clarity.
Reviewer comment #4: Clarify what is initially planted, a slip or a cutting (lie 118), as the sentence (line 141) does not make sense: “By week 2 of adding the slips, the majority grew sufficiently long enough to be considered a slip…” (suggest removing enough)
Response to comment #4: Yes, thank you for pointing this out. We have revised this sentence.
Reviewer comment #5: Line 171: Not clear what was done in triplicate (same sample read 3 times, or three different samples).
Response to comment #5: Yes, each replicate sample was measured in triplicate for the mineral and sugar analysis. This has now been made more clear.
Reviewer comment #6: Table 5 and 6 are difficult to read. Units for the different parameters need to be included, include a legend to explain the statistics.
Response to comment #6: Thank you for pointing this out. The unit of expression has now been added in both tables.
Reviewer comment #7:
Remove Clarke et al. 2002 (line 38) (using numbering system i.e., 3)
Remove Tedesco et al. 2023 (line 263)
Response to comment #7: These have now been fixed.
Author Response
Reviewer #2
This manuscript “Type of black soldier fly (Hermetia illucens) frass on sweet potato slip (Ipomea batatas) production in an aquaponic system” contains some interesting data. However, here are some of my concerns with respect to the current version of the manuscript:
Reviewer comment #1: The title is too long and must be revised.
Response to comment #1: We have modified the title according to reviewer #1. Not sure how to shorten this further while conveying what the study was about.
Reviewer comment #2: The hypothesis is not well explained in the abstract.
Response to comment #2: We have now included a hypothesis.
Reviewer comment #3: Avoid the use of abbreviations in the abstract.
Response to comment #3: When using abbreviations in the abstract, we defined them first and then the abbreviation was used. This is standard practice in manuscripts, particularly in abstracts where wording needs to be concise.
Reviewer comment #4: In lines 38, 121, and 263, the citation is inconsistent with the journal format.
Response to comment #4: Thank you for pointing these out. These have been fixed.
Reviewer comment #5: Line 49-50 has repetition and should be revised.
Response to comment #5: Thank you, this has now been fixed.
Reviewer comment #6: Line 96, define AOAC.
Response to comment #6: This has now been defined in the Materials and Methods section.
Reviewer comment #7: The aquaponics system remains the same throughout the experiment or changed.
Response to comment #7: Aquaponic system was the same throughout the study.
Reviewer comment #8: Why the mean values in the tables don’t have the letters for statistical differentiations?
Response to comment #8: There are several ways to illustrate significant differences. Because this was done with a 2-way ANOVA, where the main effects are time and frass type, the statistical differential was used for these main effects and not each individual data point.
Reviewer comment #9: Line 346 has some punctuation errors.
Response to comment #9: The semicolons have now been replaced with commas.
Reviewer comment #10: Check the whole manuscript for spelling, spacing, and grammatical mistakes.
Response to comment #10: We have gone through and have noticed areas that could be polished or made clearer, which has now been done. We would like to highlight that the first draft of this manuscript was written by a native English speaker/writer. This was further edited by research scientists at USDA-ARS (Carl Webster and Steven D. Rawles) who are also native English speakers/writers. Nevertheless, we have gone through again to polish the manuscript including providing some minor edits. If there are any specific cases of grammatical errors now, we would be grateful if these can be pointed out.
Reviewer 3 Report
The manuscript is very well drafted. There is only one minor remark: The number of replicates is missing in the headers of the tables. Please add it.
The English style is very good.
Author Response
Reviewer #3
Reviewer comment #1: The manuscript is very well drafted. There is only one minor remark: The number of replicates is missing in the headers of the tables. Please add it.
Response to comment #1: Thank you for your encouraging comments. We have now added the number of replicates.
Round 2
Reviewer 1 Report
Accept revised manuscript.
Author Response
Thank you for reviewing and accepting our revised manuscript
Reviewer 2 Report
1) The title is same as before and must be revised.
2) There is no hypothesis in the abstract as well as in the introduction.
3) Avoid the use of abbreviations in the abstract.
4) Revise the citations according to the journal format.
5) I am still confused about the analysis. The answer provided by the author is not satisfactory. There should be a clear indication that the treatments are significance.

English can be improved.
Author Response
Dear Editor,
Thank you for assessing our manuscript (Horticulturae 2603481). Two out of the three reviewers have recommended accepted, while one of the reviewers would like to see some additional changes/explanations.
We have copied/pasted the reviewer comments below along with our point-by-point response to each one. We hope that this revised manuscript is improved over the last one and our answers adequately address the reviewer’s question regarding statistical analysis.
We sincerely thank you for your kind consideration of our work,
Sincerely,
Nicholas Romano (Ph.D.)
Associate Professor / Extension Specialist & Coordinator
Virginia Cooperative Extension
Virginia State University
Reviewer comment 1) The title is same as before and must be revised.
Response to comment 1) We have shortened the title so it fits onto two lines (instead of three). Hoping this is OK. If not, perhaps the reviewer can kindly suggest a title that is shorter that conveys the type of study.
Reviewer comment 2) There is no hypothesis in the abstract as well as in the introduction.
Response to comment 2) My apologies, this has now been added.
Reviewer comment 3) Avoid the use of abbreviations in the abstract.
Response to comment 3) All abbreviations in the abstract have now been removed.
Reviewer comment 4) Revise the citations according to the journal format.
Response to comment 4) Thank you, this has been done.
Reviewer comment 5) I am still confused about the analysis. The answer provided by the author is not satisfactory. There should be a clear indication that the treatments are significance.
Response to comment 5) The proper method of statistically analyzing the data regarding sweet potato slip composition and performance in response to harvest time and frass type is to use a factorial mixed model. Thus, data in Tables 5 and 6 only requires a significance denotation if the main effects (frass type and/or harvest period) are significant. We also analyzed for their interaction, but interactions were not significantly different.
In Table 5, only harvest period was significantly different, which we denoted treatment differences by using different letters next to the means by variable analyzed (column data).
Likewise, in Table 6, only the main effect of “time” was significantly different among the variables analyzed and we denoted significance by using different letters next to the mean values.
Reviewer 3 Report
The manuscript is now very good!
Author Response
Thank you for accepting our revised manuscript!
Round 3
Reviewer 2 Report
I have no more comments.